# Brazil Nut Semi-Defatted Flour Oil: Impact of Extraction Using Pressurized Solvents on Lipid Profile, Bioactive Compounds Composition, and Oxidative Stability

**DOI:** 10.3390/plants13192678

**Published:** 2024-09-24

**Authors:** Karen Keli Barbosa Abrantes, Tatiana Colombo Pimentel, Camila da Silva, Oscar de Oliveira Santos Junior, Carlos Eduardo Barão, Lucio Cardozo-Filho

**Affiliations:** 1Programa de Pós-Graduação em Engenharia Química, Universidade Estadual de Maringá, Av. Colombo, 5790, Maringá 87020-900, PR, Brazil; kkelibarbosa@gmail.com (K.K.B.A.); csilva@uem.br (C.d.S.); lcfilho@uem.br (L.C.-F.); 2Instituto Federal do Paraná (IFPR), Rua José Felipe Tequinha, 1400, Paranavaí 87703-536, PR, Brazil; tatiana.pimentel@ifpr.edu.br; 3Laboratório de Química de Alimentos, Programa de Pós-Graduação em Química, Universidade Estadual de Maringá (UEM), Av. Colombo, 5790, Maringá 87020-900, PR, Brazil; oliveirasantos.oscardeoliveira@gmail.com

**Keywords:** pressurized extraction, phytosterols, circular economy, *Bertholletia excelsa* S.B.H

## Abstract

Brazilian nuts are native to the Amazon rainforest and are considered a non-timber forest-product of extreme economic importance to local populations. This study evaluated the lipid profile, bioactive compounds, and oxidative stability of semi-defatted Brazilian nut flour oil (BNSDFO) obtained using pressurized fluids (*n*-propane at 40 °C and 2, 4, and 8 MPa or a CO_2_/*n*-propane mixture at 40 °C and 12 MPa). A Brazilian nut kernel oil (BNKO) processed by conventional cold pressing was also obtained. The BNKO showed a higher concentration of total phenolic compounds and saturated fatty acids, higher antioxidant activity, and the presence of gallic acid derivatives. The oils extracted using pressurized fluids showed a higher concentration of linoleic acid, β-sitosterol, and polyunsaturated fatty acids. The utilization of pressurized *n*-propane resulted in higher yields (13.7 wt%), and at intermediate pressures (4 MPa), the product showed myricetin 3-*O*-rhamnoside and higher oxidative stability (OSI, 12 h) than at lower pressures (2 MPa). The CO_2_/n-propane mixture of pressurized solvents resulted in higher concentrations of squalene (4.5 times), the presence of different phenolic compounds, and a high OSI (12 h) but lower yield (2.2 wt%). In conclusion, oils with better fatty acid profiles (oleic e linoleic acids), phytosterol composition, and suitable radical scavenging activity may be obtained using pressurized fluids and Brazilian nut flour, a byproduct of oil extraction. The mixture of solvents may improve the concentration of squalene, whereas using only *n*-propane may increase oil yield.

## 1. Introduction

The Brazil nut tree (*Bertholletia excelsa* S.B.H.) originates from the Brazilian Amazon. The Brazil nut is considered one of the most important non-timber forest products globally, and 28,000 tons were produced in 2022/2023, mainly in Bolivia (79%) [1]. It is rich in micronutrients such as selenium, magnesium, and copper. Therefore, it is promoted as part of a healthy diet in several countries, being a primary source of selenium and a vital antioxidant for the modulation of the immune system and the regulation of thyroid hormones [2,3,4,5]. Furthermore, its nutritional properties include a high concentration of polyphenols and a lipid composition comprising 75% polyunsaturated fatty acids [6,7]. They are collected in conservation units by traditional communities, which combine nut processing with the conservation of native forests. These units represent a crucial source of income through the sale of nuts and their oil, which is widely used in the cosmetic industry [8,9,10].

Several methods can be used to extract Brazil nut oil, including solid–liquid extraction with petroleum ether, ethanol [11], carbon dioxide [7], supercritical fluids, and cold pressing. The latter is the most common on an industrial scale due to its reduced cost. However, it has limited efficiency and may degrade the bioactive components. Consequently, nut oil extraction by cold pressing generates semi-defatted flour as a byproduct, which still contains oil in its composition [10].

The primary and recent advancements in the extraction of oil [10,12,13,14] and bioactive compounds [15,16,17] from Brazil nuts are associated with pressurized fluid technology (ethanol, isopropyl alcohol, CO_2_, *n*-propane), with particular emphasis on process intensification, especially ultrasound-assisted techniques. The pressurized liquid extraction (PLE) technique, combined with green solvents, stands out in the extraction of oils and bioactive compounds. The low thermal degradation of the extracted bioactive compounds justifies pressurized fluid technology [18].

Extraction with pressurized fluids has emerged as a promising alternative, presenting the ability to operate under conditions that preserve the nutritional and bioactive quality of the oil seeds, including BNKO and linseed (*Linum usitatissimum*) as well as allowing for the simultaneous extraction of sunflower and olive oils [10,13,19]. This method is compared to hexane, one of the most widely used solvents for extracting vegetable oils, which can leave residues in food and is highly toxic to the reproductive and nervous systems [20]. *n*-propane has been recommended as a pressurized fluid for vegetable oil extraction, as it has high solubility in these matrices, requires shorter extraction times, and uses small amounts of solvents [21]. The principal disadvantage of using n-propane with an extraction solvent is its flammability [22]. Furthermore, utilizing solvent mixtures may result in vegetable oils of superior quality, with CO_2_ as a suitable option to be associated with *n*-propane because it is chemically inactive, non-toxic, economical, approved as a food-grade solvent, and easily separated from extracts [23]. However, the performance of mixtures of pressurized fluids in extracting vegetable oils, mainly from Brazil nuts, is understudied.

The processing parameters, particularly pressure, are paramount in extractions with pressurized fluids [18]. Therefore, this study evaluated the nutraceutical composition (lipid profile, oxidative stability, radical scavenging activity, content, and composition of phenolic compounds) of oils obtained from semi-defatted Brazil nut flour, a byproduct of the oil extraction process by cold pressing, with high nutritional relevance. The performance was evaluated under different pressure conditions. Oil was also obtained from the Brazil nut kernel by conventional cold pressing. Pressurized *n*-propane and a mixture of *n*-propane and CO_2_ were used to preserve the nutritional properties and lipophilic components and improve bioactive compounds’ availability without toxic solvents. In addition, principal component analysis was used to correlate the chemical composition with the extracted oils’ radical scavenging activity and total phenolic content. The novelty of the work is related to the high nutritional quality of the oil, extracted with pressurized solvents (n-propane and a mixture of CO_2_/*n*-propane) without the presence of solvent residue from the semi-fatted Brazil nut flour, allowing value to be added to this by-product.

## 2. Results and Discussion

### 2.1. Extraction Yields

Table 1 shows the oil yields for the different extractions. The oil extraction from BNSDFO yielded 13.7–13.8 wt% using pressurized *n*-propane as the solvent.

Previous studies have reported that oil yield from Brazil nut kernels is 22–40%, depending on temperature [7,10]. In this way, we could observe a 50% reduction in oil yield from kernel to semi-defatted flour, which may be considered satisfactory as a byproduct with reduced fat content was used [11].

The results indicate similar yields of OPF[p] of 13.7–13.8 wt%, regardless of the pressure conditions used (*p* > 0.05). Therefore, the pressure did not affect the oil yields (*p* > 0.05). Pressure increases may improve the oil yields when the temperature is close to the critical temperatures (96 °C for n-propane), improving the solvation power and oil extraction yield [24]. The lower temperatures in the present study (40 °C) may have limited the pressure impact. These results suggest that low and intermediate pressure conditions (40 °C and 2 MPa, OPF[p3] and 4 MPa, OPF[p2]) may be the most efficient extractions using pressurized *n*-propane, considering the lower energy consumption involved.

In contrast, OPF[m] showed a significantly decreased average mass yield (2.2 wt%), which was up to six times lower than OPF[p]. *n*-propane is considered a more efficient solvent for rapid extractions due to its greater oil recovery capacity and lower solvent/solid ratio [25]. The addition of CO_2_ may have caused instability in the liquid phase, limiting the miscibility of pressurized fluids and oil [26]. Regarding oil yield, using pressurized *n*-propane would be better than a mixture with CO_2_. However, despite yields, the composition of vegetable oils is paramount.

### 2.2. Fatty Acid Profile and Bioactive Compounds

As shown in Table 2, the predominant fatty acids in all analyzed oils (BNKO, OPF[p1], OPF[p2], OPF[p3], OPF[m]) were oleic acid (30–33%), a monounsaturated fatty acid (MUFA), and linoleic acid (38–42%), a polyunsaturated fatty acid (PUFA).

These findings agree with previous studies on the chemical composition of oil extracted from Brazil nuts [10,22,27].

The oil obtained by cold pressing (BNKO) showed a higher concentration of stearic acid, with a consequent increase in saturated fatty acid (SFA) (*p* < 0.05). On the contrary, the oils obtained using pressurized fluids (OPF[p1], OPF[p2], OPF[p3], OPF[m]) showed a higher concentration of linoleic acid, consequently increasing PUFA levels (*p* < 0.05). Linoleic acid is an essential fatty acid, as the human body does not synthesize it. It may reduce the risk of several diseases, such as cancer and cardiac diseases [27]. The pressurized fluids may have increased the solvent affinity for linoleic acid and improved the solvation power compared to cold pressing [28,29].

The utilization of a mixture of solvents contributed to the increase in palmitic acid (*p* < 0.05) but without impact on SFA (*p* > 0.05). Finally, the utilization of lower pressures in *n*-propane extractions (OPF[p2], OPF[p3]) resulted in lower concentrations of heneicosanoic acid (*p* < 0.05), but without impact on SFA (*p* > 0.05). Therefore, our results suggest that pressurized fluids contributed to a better fatty acid profile, regardless of pressures or solvents used, mainly due to higher linoleic acid concentrations. Although changes in pressure and solvent type may impact specific saturated fatty acids, the overall SFA content was not impaired.

Concerning active compounds (CAs), the oils showed squalene in greater quantities (343.56–1007.44 mg 100 g^−1^ oil). Furthermore, the oils presented β-sitosterol levels ranging from 26 to 40 mg 100 g^−1^ of oil and (β + γ)-tocopherol levels ranging from 16.30 to 16.88 mg 100 g^−1^ of oil (Table 2), corroborating previous studies [30,31]. The oils obtained using pressurized fluids (OPF[p1], OPF[p2], OPF[p3], OPF[m]) showed a higher concentration of β-sitosterol (*p* < 0.05). The results suggest that pressurized fluids facilitated the recovery of β-sitosterol, and the cold pressing may have resulted in some degradation [32]. A higher concentration of plant sterols in vegetable oils is desired due to the recognized health properties of these compounds, mainly to reduce blood cholesterol [26,28].

The utilization of a mixture of solvents contributed to the increase in squalene content (*p* < 0.05); therefore, OPF[m] oil was approximately 4.5 times higher in squalene content than BNKO. Thus, using a CO_2_/*n*-propane solvent mixture effectively extracted squalene (1007 mg 100 g^−1^ of oil), which stands out from other methods that used water and presented lower squalene levels [12,33]. The mixture of solvents may have provided attractive properties in terms of solvation power, resulting in a higher affinity for the substrate (squalene) compared to *n*-propane alone [26].

Finally, the utilization of lower pressures in *n*-propane extractions (OPF[p3]) resulted in lower concentrations of racemic mixture tocopherol (*p* < 0.05). The utilization of intermediate pressures maintained the racemic mixture tocopherol content (*p* > 0.05). In this way, the lower pressures may not be able to extract this bioactive compound from the matrix [29].

Table 3 shows the oils’ TPC content and DPPH radical scavenging activity. The oils showed 5.14–8.23 mg EAG 100 g^−1^ oil of TPC and 197.45–366.15 µmol 100 g^−1^ oil of DPPH values.

Studies indicate that the antioxidant content in vegetable oils is generally not high [21]. DPPH values for oils obtained from Brazil nuts with acetone were reported to be 390 µmol 100 g^−1^ of extract [33] (John and Shahidi, 2010), like the value observed in this study for BNKO (366 µmol 100 g^−1^ of oil).

The oil obtained by cold pressing (BNKO) showed a higher concentration of total phenolic compounds, with a consequent increase in the radical scavenging activity by DPPH (*p* < 0.05). In this way, the radical scavenging activity of BNKO was approximately 1.8 times higher than that of oils extracted from BNSDFO with different pressurized solvents. The phenolic compounds are generally more concentrated in the outer parts of the kernel [34], which explains the higher TPC values in BNKO compared to BNSDFO.

All the oils obtained with pressurized fluids showed similar TPC content and DPPH values (*p* > 0.05), suggesting that different pressures and solvent mixtures did not impact these parameters. Phenolic compounds are known for their ability to capture free radicals and are associated with disease prevention. The literature suggests that the solubility of these compounds varies depending on the solvent used in the extraction, directly reflecting the observed radical scavenging activity [35]. However, in our study, the different pressures and solvent composition did not affect the oils’ TPC content and radical scavenging activity.

The two main dimensions of principal component analysis (PCA) summarize 90.9% of the variance in chemical composition, including fatty acids (FAs), active compounds (ACs), radical scavenging activity, and TPC content of the oils (Figure 1). The first principal component (PC1) explains 66.5%, and the second principal component (PC2) explains 24.4%. The most significant contributions to PC1 were PUFA (9.82%), MUFA (9.7%), total phenolic compounds (9.79%), and antioxidant capacity measured by the DPPH test (9.71%). In PC2, palmitic acid (24.79%) and stearic acid (26.46%) were the most representative.

PC1 differentiated the pressed oil from BNKO, located on the positive axis, from the oils from BNSDFO. In this way, BNKO was characterized by a higher total phenolic compound content and more significant radical scavenging activity but higher SFA content. On the contrary, oils extracted with pressurized fluids were characterized by higher PUFA levels and higher concentrations of linoleic acid and β-sitosterol.

PC2 differentiated the oils obtained using pressurized fluids. Higher concentrations of squalene and palmitic acid characterized the oils obtained with the mixture of solvents located above the axis. The PCA results indicate that BNSDFO extraction using a mixture of CO_2_ and *n*-propane was particularly effective for obtaining oils rich in squalene. Furthermore, extraction with pressurized fluids proved a viable option for obtaining oils with high levels of PUFA, especially linoleic acid.

The type of phenolic compound is essential, in addition to the total content, when evaluating the phenolic compounds. Phenolic analysis of oils identified 24 compounds, including phenolics and flavonoids (Table 4). Hydroxybenzoic acid derivatives such as gallic acid, 4-hydroxybenzoic acid, and ellagic acid stand out among the phenolics. Furthermore, all oils showed ferulic acid, protocatechuic acid derivative, succinic acid, vanillin, catechin, epicatechin, catechin gallate, epicatechin gallate, quercetin, and taxifolin. Similar phenolic compounds were reported by [34,36].

The oil obtained by cold pressing (BNKO) showed the gallic acid derivative as an exclusive compound. Utilizing lower pressures (OPF[p2] and OPF[p3]) resulted in identifying p-coumaric acid, protocatechuic acid, ellagic acid derivative, and pyruvic acid. Utilizing higher pressures (OPF[p1]) resulted in the identification of vanillic acid, ascorbic acid, epigallocatechin 3-ogallate, and myricetin-3-*O*-rhamnoside. Finally, a mixture of solvents (OPF[m]) identified ellagic acid derivative, ascorbic acid, and myricetin-3-*O*-rhamnoside. Although not a phenolic compound, the presence of citric acid in oils may be related to its potential role as a metal chelator, given the high concentration of selenium in Brazil nuts, suggesting a synergy between the phenolic components and the mineral context of the matrix [34]. Our results demonstrate that pressure and solvent composition may change the phenolic composition. Some identified compounds are known for their antioxidant, antimicrobial, anti-inflammatory, anticancer, antiplatelet aggregation, anxiolytic, and analgesic properties and beneficial effects against diabetes, obesity, hyperglycemia, and gout [37,38].

### 2.3. Oxidative Stability (OSI) of Oils

The oils’ oxidative stability (OSI) is influenced by factors such as the degree of unsaturation of fatty acids, the location of double bonds in the carbon chain, the process temperature, and the presence of oxygen [39]. A significant variation in OSI values was observed between the different pressurized solvents and the working pressures used. Notably, OPF[p1], OPF[p2], and OPF[m] oils demonstrated the highest oxidative stability, reaching 12 h (Table 5), whereas OPF[p3] showed the lowest stability at 6.5 h. The lower OSI of OPF[p3] may be related to the lower racemic mixture tocopherol content because this compound protects the lipids from oxidation reactions [26,32].

Comparatively, the oils under the evaluated conditions exhibited oxidative stability comparable or superior to that of other oilseeds such as almonds (11–14 h), pecans (8.5–10.8 h), pistachios (15–18 h) and walnuts (2.5–3.5 h) [40]. Furthermore, Brazil nuts extracted using supercritical carbon dioxide recorded an oxidation induction time of 14.85 h [7].

## 3. Materials and Methods

### 3.1. Materials and Cold Pressing Oil Processing

The Cooperativa dos Agricultores do Vale do Amanhecer (COOPAVAM) supplied the Brazil nuts (Juruena/MT, Brazil, Lot 72, produced in June 2023). The nuts were first heated to 65 °C and subsequently subjected to pressing. After extraction, the oil was filtered to remove small nut fragments, resulting in the Brazil nut kernel oil (BNKO). The resulting flour, still containing residual oil, was used for extraction with pressurized solvents, and the product was denominated Brazil nut semi-defatted flour oil (BNSDFO).

### 3.2. Pressurized Solvent Extraction

*n*-Propane (Linde, 95% purity) and inert carbon dioxide (Pluma Gases LTDA) were used as solvents for extractions using pressurized fluids. Extractions using *n*-propane and the mixture of CO_2_ with *n*-propane were performed in duplicate using the experimental apparatus described by Trentini et al. (2017) [16]. For that, 110 g of Brazil nut semi-defatted flour was used in an extraction container with a volume of 165.2 cm^3^.

Initially, a preliminary test with *n*-propane determined the operating temperature, ranging from 30 to 50 °C, with a midpoint of 40 °C. Due to the lack of significant differences in yields between the temperatures tested and the importance of maintaining temperatures below 70 °C to avoid the degradation of thermolabile bioactive compounds, it was decided to continue the experiments at 40 °C. The pressures used were 8 MPa (OPF[p1]), 4 (OPF[p2]), and 2 (OPF[p3]), considering the advantages of *n*-propane in operating at lower pressures than other solvents, such as carbon dioxide, which represents a significant advantage for industrial applications [41]. The extraction time was 50 min, with a constant solvent flow rate of 5 mL min^−1^. The CO_2_/*n*-propane solvent mixture experiment (OPF[m]) was carried out at 40 °C and pressures of 12 MPa, with an extraction time of 100 min and 5 mL min^−1^ of solvent flow rate. The increased pressures and times were needed due to the solvent mixture and determined in preliminary tests.

The extracted oils were collected in amber bottles, and the extraction kinetics were monitored by periodic weighing using an analytical balance (model APX-200, Denver Instrument). For extractions with pressurized *n*-propane, weighing occurred every 5 min, and for solvent mixtures, every 10 min. According to Equation (1), the percentage mass yield was calculated by the ratio between the mass of oil extracted and the mass of flour used.
(1)$$Y\left(wt\%\right)=\frac{M}{Gr}\cdot 100$$
where Y is the percentage yield by mass, M is the mass of oil accumulated during extraction in grams, and Gr is the used mass of flour on a dry basis.

### 3.3. Characterization of the Oils

The oils were characterized in triplicate, regarding their fatty acid profile, the total phenolic compound content (TPC) and phenolic type, phytosterol content, tocopherol content, oxidative stability evaluated by the Rancimat method, and radical scavenging activity using the 2,2-diphenyl-1-picryl-hydrazyl-hydrate (DPPH) assay.

#### 3.3.1. Fatty Acids, Phytosterols and Tocopherols Analysis by GC/MS

The composition of fatty acids (FA) and the content of active compounds (AC) in the BKNO and BNSDFO were analyzed using gas chromatography coupled with mass spectrometry (GC-MS, Shimadzu GC-MS QP2010 SE). For the analyses, helium was used as carrier gas at a 1.0 mL/min flow rate, fractionation rate of 1:40, and injection volume of 2 µL. The injection and GC-MS interface temperatures were set at 250 °C and 280 °C, respectively, and the ion source temperature was 260 °C. Mass spectra were acquired at 70 eV, covering a range from 50 to 550 m/z. For identification, the mass spectra of the compounds detected were compared with those of the NIST 14 library, the Pubchem database established by the National Library of Medicine, and the NIST standard reference database number 69 (NIST Chemistry WebBook).

Samples were saponified and derivatized using 14% BF3 in methanol for methylation to determine fatty acids [42]. Dilution was carried out with heptane, and a ZB-Wax TM capillary column (Zebron, 30 m × 0.25 mm × 0.25 μm) was used. The chromatographic oven was initially adjusted to 80 °C, with a heating rate of 10 °C min^−1^ up to 240 °C, maintaining this temperature for 2 min. The percentage of each compound was determined using the area normalization method, which is defined as the ratio between the area of the individual compound and the total area of the peaks obtained in the chromatogram.

For the analysis of phytosterols and tocopherols, the samples were derivatized with N, O-Bis(trimethylsilyl)trifluoroacetamide with trimethylchlorosilane (BSTFA/TMCS) at 60 °C for 30 min. A total of 80 µL of 5-α-cholestane standard at 5 mg mL^−1^ was added to the derivatized samples to quantify the content of each compound, which was diluted with heptane to a final concentration of 40 mg mL^−1^. The injection was carried out on an SH-RTx-5MS capillary column (Shimadzu, 30 m × 0.25 mm × 0.25 µm), following a heating ramp identical to that used for other vegetable oils [22,30,32].

#### 3.3.2. Determination of Total Phenolic Content and DPPH Assay

Before the analyses, 500 mg of oil was dissolved in 1.5 mL of *n*-hexane and then extracted with methanol (3 × 1 mL with stirring for 2 min), and the mixture was incubated for 16 h in the dark [43]. The methanolic extract was used to quantify the TPC content and radical scavenging activity by DPPH assay, as described by [44]. The absorbance of the samples was measured using a UV spectrophotometer (Shimadzu, UV-1900, Tokyo, Japan). TPC content was quantified using a gallic acid standard curve (GAE) and expressed in mg per g of dry extract. To quantify radical scavenging activity, a calibration curve was prepared using Trolox as a standard (R^2^ > 0.99), and the results were expressed as μmol of Trolox-equivalent antioxidant capacity (TEAC) per 100 g of oil.

#### 3.3.3. UHPLC-ESI-MS Profile of Phenolic Compounds

The detection of phenolic compounds was carried out using an ultra-high performance liquid chromatograph (UHPLC) coupled to a triple quadrupole mass spectrometer (MS/MS), model Waters ZTQD LCMS Acquity, equipped with an electrospray ionization (ESI) source operating in positive ionization mode. Data were acquired and analyzed using MassLynx software version number 4.1, and the Pubchem database was used to identify compounds. Multiple reaction monitoring (MRM) mode was used to determine the analytes. The samples were subjected to a flow rate of 20 µL min^−1^ and diluted in a ratio of 100 µL of sample to 900 µL of HPLC grade methanol. A 0.1% formic acid solution was added to the mixture. Operating conditions included a capillary voltage of 3.0 kV, a cone of 30 V, a desolvation temperature of 400 °C, and a nebulizer gas (nitrogen) flow of 600 L/min.

#### 3.3.4. Oxidative Stability Analysis by Rancimat

The induction period (IP), using the Rancimat 743 equipment (Metrohm, Herisau, Switzerland), was used to evaluate the oxidative stability of oil samples under accelerated conditions. For analyses, conductivity cells were filled with Mili-Q water, and 3.0 g of the oil samples were exposed to saturated air at a flow rate of 20 Lh^−1^ at a temperature of 110 °C. The variation in water conductivity over time indicated the formation of volatile compounds. The abrupt increase in conductivity, identified from the corresponding graph, was marked as the beginning of oxidative degradation or the rancidity point, as described by [45].

### 3.4. Statistical Analysis

The data were subjected to analysis of variance (ANOVA) and Tukey’s test for mean comparisons (*p* < 0.05) using the SISVAR software version 5.7 [46]. Additionally, principal component analysis (PCA) was applied to simplify the set of variables and evaluate the influence of fatty acids (FA) and active compounds (AC) on the results obtained from BNKO, OPF[p]s and OPF[m], using the FactoMinerR software in the R Studio version number 4.3.0 Inc. (Boston, MA, USA environment) [46].

## 4. Conclusions

This study highlighted variations in the chemical composition of oils extracted from semi-defatted Brazil nut flour (BNSDFO) using different pressurized solvents compared to the traditional cold-pressing extraction method. The Brazil nut oil from cold pressing showed a higher TPC concentration and saturated fatty acids, greater radical scavenging activity, and gallic acid derivatives. The oils extracted using pressurized fluids showed a better fatty acid (higher percentage of linoleic acid and PUFA) and phytosterol (β-sitosterol) composition. The utilization of pressurized *n*-propane resulted in higher yields (13.7 wt%), and it is suggested to use intermediate pressures (OPF[p2], 4 MPa) due to the phenolic composition (myricetin 3-*O*-rhamnoside) higher oxidative stability and maintenance of phytosterol composition. The utilization of a mixture of pressurized solvents resulted in higher concentrations of squalene (4.5 times), the presence of different phenolic compounds (ellagic acid derivative and myricetin 3-*O*-rhamnoside), and high OSI (12 h), but lower yield (2.2 wt%). In conclusion, the pressurized fluid extraction technology applied to Brazil nuts and semi-fatted Brazil nut flour has yielded oils with higher percentages of linoleic acid and β-sitosterol content than cold-press extraction. Using a solvent mixture comprising CO_2_ and *n*-propane did not alter the lipid profile and significantly recovered the squalene content present in the oil. This solvent combination reduces the risk of explosion and aligns with the concept of a green solvent, demonstrating carbon dioxide’s affinity for squalene.

## Figures and Tables

**Figure 1 plants-13-02678-f001:**
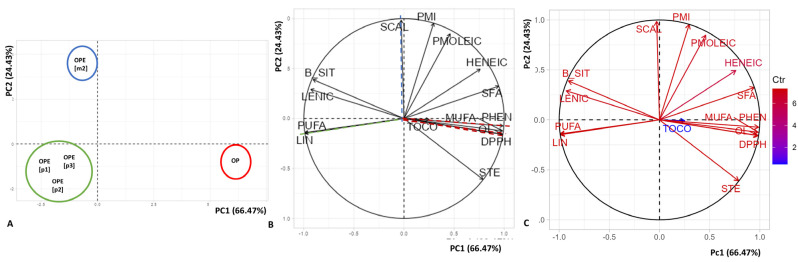
(**A**,**B**) Principal component analysis (PCA) relating the first and second principal components (PC1 and PC2) relating to the composition of fatty acids and minor compounds comparing the solvent used and the extraction method; (**C**) Loading plot to describe the fatty acids and minor compounds giving the most contributing to PCA; PMl: palmitic acid; PMOLEIC: palmitoleic acid; STE: stearic acid; Ol: oleic acid; LIN: linoleic acid; LENIC: linolenic acid; HENEIC: heneicosanoic acid; SCAL: squalene; TOCO: (β + γ)-tocopherol. B.SIT: β-Sitosterol; SFA: saturated fatty acids; MUFA: monounsaturated fatty acids; PUFA: polyunsaturated fatty acids; PHEN: phenolics total; DPPH: radical scavenging activity for DPPH.

**Table 1 plants-13-02678-t001:** Experimental conditions and extraction yield of oils.

Run	Code	Solvent	Matrix	Time (min) ^a^	T (°C)	P (MPa)	Oil Yield (wt%)
1	OPF[p1]	*n*-propane	BNSDFO	50	40	8	13.7
2	OPF[p2]				40	4	13.8
3	OPF[p3]				40	2	13.7 ± 0.3 ^b^
4	OPF[m]	CO_2_ + *n*-propane *	BNSDFO	100	40	12	2.2
5	BNKO	Cold pressing	BNKO **				31.3 ± 3.0 ^b^

^a^ Time of the dynamic extraction; ^b^ Average value ± standard deviation of two replicates. * Solvent ratio 40% CO_2_/60% *n*-propane **. BNKO = Brazil nut kernel oil from cold pressing; OPF-p = extraction with pressurized fluid—*n*-propane at 8 [p1], 4 [p2], and 2 [p3] MPa; OPF-m = extraction with pressurized fluid—mixture (CO_2_/*n*-propane).

**Table 2 plants-13-02678-t002:** Profile of fatty acids and active compounds of oils using different solvents and extraction parameters.

Fatty Acid (%)	[M+] (%)	Fragment Ions (m/z) *	BNKO	OPF[p1]	OPF[p2]	OPF[p3]	OPF[m]
Palmitic (C16:0)	270	74, 87, 55, 75, 57, 143, 69, 59, 83, 227	15.2 ± 0.0 ^b^	14.7 ± 0.2 ^b^	14.6 ± 0.0 ^b^	14.8 ± 0.1 ^b^	16.07 ± 0.0 ^a^
Palmitoleic (C16:1)	268	55, 69, 74, 83, 87, 67, 84, 81, 97, 96	0.3 ± 0.0 ^a^	0.3 ± 0.0 ^a^	0.3 ± 0.0 ^a^	0.3 ± 0.0 ^a^	0.3 ± 0.0 ^a^
Stearic (C18:0)	298	74, 87, 55, 75, 57, 143, 69, 83, 59, 255	13.3 ± 0.1 ^a^	12.3 ± 0.0 ^c^	12.4 ± 0.0 ^a,b^	12.3 ± 0.0 ^c^	11.7 ± 0.0 ^d^
Oleic (C18:1)	296	55, 69, 74, 83, 97, 84, 87, 96, 67, 81	31.3 ± 0.0 ^a^	30.0 ± 0.2 ^a^	30.3 ± 0.1 ^a^	30.2 ± 0.1 ^a^	30.2 ± 0.1 ^a^
Linoleic (C18:2)	294	67, 81, 55, 95, 68, 82, 79, 96, 69, 109	39.4 ± 0.1 ^b^	42.3 ± 0.0 ^a^	42.2 ± 0.1 ^a^	42.0 ± 0.1 ^a^	41.3 ± 0.1 ^a^
Linolenic (C18:3)	292	79, 67, 95, 93, 55, 80, 81, 108, 107, 77	0.1 ± 0.0 ^b^	0.1 ± 0.0 ^a^	0.1 ± 0.0 ^a^	0.1 ± 0.0 ^a^	0.1 ± 0.0 ^a^
Heneicosanoic (C21:0)	326	74, 87, 55, 57, 75, 69, 143, 83, 59, 97	0.3 ± 0.0 ^b^	0.3 ± 0.1 ^a,b^	0.2 ± 0.0 ^a^	0.2 ± 0.0 ^b^	0.3 ± 0.0 ^b^
SFA			28.9 ^a^	27.3 ^c^	27.2 ^c^	27.4 ^b,c^	28.0 ^b^
MUFA			31.6 ^a^	30.3 ^a^	30.6 ^a^	30.5 ^a^	30.5 ^a^
PUFA			39.5 ^b^	42.5 ^a^	42.3 ^a^	42.1 ^a^	41.4 ^a^
**Bioactive compounds (mg 100 g^−1^ oil)**							
Squalene	410	69, 81, 95, 68, 67, 121, 93	392.60 ± 3.46 ^b^	343.56 ± 2.32 ^c^	345.88 ± 4.35 ^c^	354.88 ± 1.14 ^c^	1007.44 ± 3.96 ^a^
(β+γ)-tocopherol	488	488, 223, 73, 222, 489, 224	16.62 ± 0.10 ^a^	16.48 ± 0.15 ^a^	16.88 ± 0.09 ^a^	14.42 ± 1.20 ^b^	16.30 ± 0.25 ^a^
β-Sitosterol	486	129, 73, 57, 357, 95, 75, 81, 396	36 ± 2.57 ^b^	40 ± 1.87 ^a^	40 ± 0.55 ^a^	40 ± 3.05 ^a^	41 ± 0.69 ^a^

^*^ Fragments listed are those from the derivatized sample. For formulations and processing parameters, please see Table 1. Different superscript small letters in the same row for the same parameter denote difference (*p* < 0.05) based on the Tukey test.

**Table 3 plants-13-02678-t003:** Total phenolic compounds and radical scavenging activity (DDPH) of oils.

	Phenolics	DPPH
	mg EAG 100 g^−1^ Oil	(µmol 100 g^−1^ Oil)
BNKO	8.23 ^a^ ± 0.84	366.15 ^a^ ± 24.69
OPF[p1]	5.14 ^b^ ± 0.49	199.03 ^b^ ± 1.45
OPF[p2]	5.60 ^b^ ± 0.24	197.94 ^b^ ± 2.13
OPF[p3]	5.82 ^b^ ± 0.25	198.30 ^b^ ± 4.64
OPF[m]	5.74 ^b^ ± 0.13	197.45 ^b^ ± 2.58

For formulations and processing parameters, please see Table 1. Different superscript small letters in the same column for the same parameter denote difference (*p* < 0.05) based on the Tukey test.

**Table 4 plants-13-02678-t004:** Phenolic compounds in BNKO and BNSDFO oils.

Number	Class	Compound Annotation	Chemical Formula	[M−H]− Measured	Fragments		Occurrence			
						BNKO	OPF			
							OPF[p1]	OPF[p2]	OPF[p3]	OPF[m]
	**Phenolic acids**									
1	Hydroxybenzoic acid	Gallic acid	C_7_H_6_O_5_	169	79.02, 69.03, 51.02, 41.04,	+	+	+	+	+
2	Hydroxybenzoic acid	Gallic acid derivative	-	274	125.02, 169.01	+	−	−	−	−
3	Hydroxybenzoic acid	4-Hydroxybenzoic acid	C_7_H_6_O_3_	137.03	106.64, 93.03	+	+	+	+	+
4	Hydroxybenzoic acid	Ellagic acid	C_14_H_6_O_8_	301	257, 283, 285, 229, 184.92, 134.92	+	+	+	+	+
5	Hydroxycinnamic acids	*p*-Coumaric acid	C_9_H_8_O_3_	164.05	119.05, 91.05	−	−	+	+	−
6	Hydroxycinnamic acids	Ferulic acid	C_10_H_10_O_4_	193	133.1, 161, 177.1	+	+	+	+	+
7	Dihydroxybenzoic acid	Protocatechuic acid	C_7_H_6_O_4_	153	-	+	−	+	+	−
8	Dihydroxybenzoic acid	Protocatechuic acid derivative	-	329	153.04, 109.03, 124.03	+	+	+	+	+
9	Dihydroxybenzoic acid	Vanillic acid	C_8_H_8_O_4_	167.03	152.01, 108.02	+	+	−	−	+
10	Dihydroxybenzoic acid	Ellagic acid derivative	-	447	301, 257, 229	−	−	+	+	+
11	Dihydroxybenzoic acid	Vanillic acid derivative	-	329	167	+	+	+	+	−
12	α-Ketopropionic acid	Pyruvic acid	C_3_H_4_O_3_	87.06	59.01	+	−	+	+	+
13	Ketoaldonic Acid	Ascorbic acid	C_6_H_8_O_6_	175.03	147.2, 87.00, 69.03	−	+	−	−	+
14	Dicarboxylic Acid	Succinic acid	C_4_H_6_O_4_	117.02	72.91	+	+	+	+	+
15	Phenolic Aldehyde	Vanillin	C_8_H_8_O_3_	151.05	137.05, 123.05, 109.0, 81.0	+	+	+	+	+
	**Flavanoids**									
16	Flavanol	Catechin	C_15_H_14_O_6_	289.1	136.8, 150.7, 160.8	+	+	+	+	+
17	Flavanol	Epicatechin	C_15_H_14_O_6_	279	109.01, 121.01, 123.03, 125.01, 137.00	+	+	+	+	+
18	Flavanol	Catechin gallate	C_22_H_18_O_10_	441.03	109.01, 125.00, 168.98, 289.03	+	+	+	+	+
19	Flavanol	Epicatechin gallate	C_22_H_18_O_10_	441.19	109.08, 125, 168.98, 151.10, 203.14, 245.16, 289.16	+	+	+	+	+
20	Flavanol	Epigallocatechin 3-Ogallate	C_22_H_18_O_11_	457.3	305.6, 169.1, 125.02	+	+	−	−	+
21	Flavonol	Quercetin	C_15_H_10_O_7_	300.9	179, 151	+	+	+	+	+
22	Flavonol	Myricetin-3-*O*-rhamnoside	C_21_H_20_O_12_	463	317	−	+	−	−	+
23	Flavonol	Taxifolin (dihydroquercetin)	C_15_H_12_O_7_	303.05	285.05, 179.00, 125.03	+	+	+	+	+
24	Organic Acid	Citric acid	C_6_H_7_O_7_	191	111, 173	+	+	+	+	+

For formulations and processing parameters, please see Table 1. (+) means presence, and (−) means absence.

**Table 5 plants-13-02678-t005:** Oxidative stability in BNKO and BNSDFO oils.

	OXS (h)
BNKO	8.7 ± 0.02
OPF[p1]	12.1 ± 0.34
OPF[p2]	12.4 ± 0.24
OPF[p3]	6.5 ± 0.14
OPF[m]	12.3 ± 0.30

For formulations and processing parameters, please see Table 1.

## Data Availability

Data are contained within the article.

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
