# Peer review of "Brazil Nut Semi-Defatted Flour Oil: Impact of Extraction Using Pressurized Solvents on Lipid Profile, Bioactive Compounds Composition, and Oxidative Stability"

_plants, 2024, doi:10.3390/plants13192678_

Round 1
Reviewer 1 Report (New Reviewer)
Comments and Suggestions for Authors
Abstract
L16-17: this makes me curios about the form of phenolic compounds that can be dissolved in oil phase espeically when you mentioned later the garlic acid derivatives.
L22: what solve mixture?
L24: Define better fatty acid progiles
Introduction:
L33: what does "it" refer to? along with L34,35 is it nut or oil?
** you seem to refer it in many places..passive voice could be considered alternative.
L48-L50:the placement of the citations does not make sense may be after "of oil"?
L57: You have not defined BNKO anywhere.
The argument in L60 is okay but perhaps add a line describe the limitation of propane.
L65-67: What would be the alternative green solvent you purposing then?
Materials and method
L240 Full name before abbreviation. L242-43 Then this is not cold pressing?
3.2 You did not describe the condition for "pressurised CO2" is 12 Mpa the condition at critical points for both propane and CO2?
3.3.2 I am not convince with the phenolic content in pressed oil as well as the antioxidant assay you chosen. Remember this is the oil phase how did you expect gallic acid to be dissloved in oil? Also DPPH might not be an appropriate assay for oil extract? Make sure you mention why these were tested in the introduction.
Results and discussion:
L97-1-3: Does these mean for Run 4, the pressure is not the optimum extraction condition? Also for Run 1-3 adjusting pressure dose not seem to impact yield as well as fatty acid profile. and why the SD only showed for Run 3 and 5?
See my concern with phenolic and DPPH previously.
Overall, it seems to me that combine propane and CO2 does not improve really the quantity and quanlity of the products.
Comments on the Quality of English Language
N/a
Author Response
Review 1
Abstract
- L16-17: this makes me curios about the form of phenolic compounds that can be dissolved in oil phase espeically when you mentioned later the garlic acid derivatives.
Answer: As mentioned in Item 3.3.2, phenolic compounds were extracted from the oil phase before analysis.
- L22: what solve mixture?
Answer: The mixture referred to in the text is a solvent mixture containing 40% CO2 and 60% n-propane. The text has corrected the term to CO2/n-propane pressurized solvent mixture in L23.
- L24: Define better fatty acid progiles
Answer: The fatty acids in greater concentration in the oils obtained from the semi-fatted Brazil nut flour (oleic acid and linoleic acid) have been added to the summary in L 26.
Introduction:
- L33: what does "it" refer to? along with L34,35 is it nut or oil?
Answer: The sentence refers to the Brazil nut as the most significant non-forest timber product globally.
** you seem to refer it in many places passive voice could be considered alternative.
- L48-L50: the placement of the citations does not make sense may be after "of oil"?
Answer: The suggestion was accepted and changed in the manuscript in L50.
- L57: You have not defined BNKO anywhere.
Answer: The definition of BNKO (Brazilian nut kernel oil) was presented earlier in the abstract, in L17.
- The argument in L60 is okay but perhaps add a line describe the limitation of propane.
Answer: The principal disadvantage of using the n-propane with extraction solvent is its flammability, described in L65-66.
- L65-67: What would be the alternative green solvent you purposing then?
Answer: The proposal is to use n-propane or a mixture of solvents (CO2/n-propane). Adding CO2 to n-propane may be suitable because CO2 is chemically inactive, non-toxic, economical, approved as a food-grade solvent, and easily separated from the extracts (L63-65).
Materials and method
- L240 Full name before abbreviation. L242-43 Then this is not cold pressing?
Answer: The abbreviation COOPAVAM was placed after the full name in the L254.
- 2 You did not describe the condition for "pressurised CO2" is 12 Mpa the condition at critical points for both propane and CO2?
Answer: CO2 was not used by itself on its own as an extraction solvent only in the CO2/n-propane mixture. The pressure of 12 MPa was used for the CO2/n-propane solvent mixture.
- 3.2 I am not convince with the phenolic content in pressed oil as well as the antioxidant assay you chosen. Remember this is the oil phase how did you expect gallic acid to be dissloved in oil? Also DPPH might not be an appropriate assay for oil extract? Make sure you mention why these were tested in the introduction.
Answer: The authors appreciate the reviewer's comment and indicate that phenolic compounds were extracted from the oil phase prior to analysis, as mentioned in Item 3.3.2.
The method chosen is based on electronic transfer (Single Electron Transfer), which measures the effectiveness of the antioxidant in transferring an electron to reduce oxidizing radicals (Wright et al. 2001). The DPPH molecule is characterized as a stable free radical by virtue of the delocalization of the unpaired electron throughout the molecule. This delocalization gives this molecule a violet color, characterized by an absorption band in ethanol at about 520 nm (Molyneux, 2004). This test is based on measuring the antioxidant capacity of a given substance to sequester the DPPH radical (Figure 1), reducing it to hydrazine.

Figure 1. Radical (1) and non-radical (2) forms of DPPH.
When a certain substance that acts as a hydrogen atom donor is added to a DPPH solution, hydrazine is obtained with a simultaneous change in color from violet to pale yellow. Visually, the degree of color change is correlated with the concentration of total antioxidant capacity (Munteanu et al. 2021).
This method was selected because it is considered, from a methodological point of view, one of the easiest, most accurate, and most reproducible methods for assessing antioxidant activity. Therefore, the authors believe that it is not appropriate to add information in the introduction regarding the method.
MOLYNEUX P. 2004. The Use of Stable Free Radical Diphenylpicrylhydrazyl (DPPH) for Estimating Antioxidant Activity. Songklanakarin Journal of Science and Technology 26: 211-219.
MUNTEANU IG, & APETREI C. 2021. Analytical Methods Used in Determining Antioxidant Activity: A Review. Int J Mol Sci 22(7): 3380. https://doi.org/10.3390/ijms22073380
WRIGHT JS, JOHNSON ER, & DILABIO GA. 2001. Predicting the Activity of Phenolic Antioxidants: Theoretical Method, Analysis of Substituent Effects, and Application to Major Families of Antioxidants. J Am Chem Soc 123(6): 1173–1183. https://doi.org/10.1021/ja002455u
Results and discussion:
- L97-1-3: Does these mean for Run 4, the pressure is not the optimum extraction condition? Also for Run 1-3 adjusting pressure dose not seem to impact yield as well as fatty acid profile. and why the SD only showed for Run 3 and 5?
Answer: The results indicate similar yields of OPF[p] (13.7-13.8) wt%, regardless of the pressure conditions used (p > 0.05). Therefore, the pressure did not affect the oil yields (p > 0.05). Repeats were carried out for the best extraction conditions for each solvent (n-propane and CO2/n-propane mixture). As there was no influence of pressure on extraction yields, the lowest pressure was considered the best extraction condition.
- See my concern with phenolic and DPPH previously.
Answer: Please check previous answers.
- Overall, it seems to me that combine propane and CO2does not improve really the quantity and quality of the products.
Answer: The average mass yield was reduced six times using a CO2/n-propane mixture. However, the CO2/n-propane mixture favored the extraction of the minority compounds, mainly squalene.
Reviewer 2 Report (New Reviewer)
Comments and Suggestions for Authors
This article is well written and publishable in Plants after major revision. Some comments to be addressed by authors are:
1. Good abstract should contain: Introduction, objective, methods, results and Conclusion. In abstract, the introduction part is missing. Please prove one or two sentences in your abstract.
2. In Introduction, the main advantage of Pressurized Solvents over the other extraction methods should be explored. In addition, the successful application of this method to other edible fats and oils is recommended to be cited in Introduction.
3. The novelty of this study should be explored in Introduction part.
4. It is nice if authors use the experimental design such as response surface methodology to optimize the extraction conditions which provide high yield.
5. for DPPH assay, the rems used is radical scavenging activity not antioxidant activity. Otherwise, the authors could use the terms "antioxidant activity" if the methods used are more than two mechanism of antioxidant such as radical scavenging, lipid peroxidation inhibition, etc
6. In discussion, the authors are suggested to add the loading plot to describe which variables giving the most contributing to PCA.
Author Response
Review 2
This article is well written and publishable in Plants after major revision. Some comments to be addressed by authors are:
- Good abstract should contain: Introduction, objective, methods, results and Conclusion. In abstract, the introduction part is missing. Please prove one or two sentences in your abstract.
Answer: The sentences about the introduction were inserted in L13-14, of the abstract section.
- In Introduction, the main advantage of Pressurized Solvents over the other extraction methods should be explored. In addition, the successful application of this method to other edible fats and oils is recommended to be cited in Introduction.
Answer: L57-62 identifies the advantages of the extraction method using pressurized solvents compared to other extraction methods. L59-60 describes some matrices that have been extracted and obtained good results using pressurized solvents.
- The novelty of this study should be explored in Introduction part.
Answer: The novelty of this study was inserted in the Introduction part, in L81-84.
- It is nice if authors use the experimental design such as response surface methodology to optimize the extraction conditions which provide high yield.
Answer: The response surface methodology (RSM) is applied when the aim is to optimize a response influenced by various factors. In the case of this work, the authors, after previous studies, varied only one factor, which was the extraction pressure (2 at 8 MPa). According to the ANOVA, pressure was not a significant factor. The different pressures used did not influence the extraction yield.
- for DPPH assay, the rems used is radical scavenging activity not antioxidant activity. Otherwise, the authors could use the terms "antioxidant activity" if the methods used are more than two mechanism of antioxidant such as radical scavenging, lipid peroxidation inhibition, etc
Answer: For the DPPH assay, the term antioxidant activity has been replaced throughout the manuscript by radical scavenging activity.
- In discussion, the authors are suggested to add the loading plot to describe which variables giving the most contributing to PCA.
Answer: The loading graph to describe which fatty acids and minor compounds contributed most to PCA has been added to Figure 1.
Round 2
Reviewer 2 Report (New Reviewer)
Comments and Suggestions for Authors
the authors have addressed all of my comments. therefore, I recommend to accept this manuscript
This manuscript is a resubmission of an earlier submission. The following is a list of the peer review reports and author responses from that submission.
Round 1
Reviewer 1 Report
Comments and Suggestions for Authors
The article titled "Brazil nut semi-defatted flour oil: Impact of extraction using pressurized solvents on lipid profile, bioactive compounds composition, and oxidative stability” discusses effect of extraction method on lipid profile, bioactive compounds, and stability of oils obtained from the by-product of the Brazilian nut kernel oil pressing process. However interesting the work is, it is not without some shortcomings. In my opinion, the article is carelessly written and needs significant improvements.
Some remarks:
Authors are suggested to include a more comprehensive literature review by including recent studies and advancements in the field oil extraction from BNKO.
In the introduction, the authors mention that BNKO are rich in selenium, magnesium and copper, among others. Why was the effect of the extraction process on the content of these micronutrients not investigated?
Inconsistent citation style e.g L78, L80, L134, l327
The authors use a lot of abbreviations. They seem to get lost in this and sometimes use different versions of the same abbreviation e.g. (BNSFSO=BNSFFO=BNSDFFO?? Or EPF=OPF??). This needs to be unified and corrected.
L 15: pressure should be in MPa rather than mPa.
L 25: abbreviation used in abstract should be explained.
L58-65: It is not clear from the quoted part of the article what the novelty of the research presented is. It should be clearly stated.
L71: BNSFSO abbreviation should be explained. Does this mean the same thing as BNSFFO? (e. g. L70)
L71: In the abstract and section 3.2, the authors state that the experiments were performed at 12 MPa. It should be standardised.
L71: In Table 1, the authors use the abbreviations EPF[...] and in the text the abbreviations OPF[...]. Is this correct? If so, it should be explained in detail. If not, it should be corrected throughout the text, as this inconsistency is repeated many times. I must admit that this is very disruptive to the interpretation of the research.
L74-L76: Does this paragraph refer to the explanation of the abbreviations in Table 1? It should be in smaller font to make it clear.
L101: Table 2. Same situation with abbreviations as in Table 1. OP=BNKO? It is not clear for me.
L103-105: Repeated unnecessary explanation from L74-L76. This explanation creates more confusion than it clarifies.
L149-L150: The same situation as in the L103-L105.
L104: Table 4 is very unreadable. It should be improved.
L205-L206: The same situation as in the L103-L105.
L210: Results for OPF[p1] are not presented in Table 4.
L221-L227: Why are the results of the oxidative stability tests not presented in the form of a table or figure?
L240: BNSDFFO is used in the same meaning as BNSFFO?
Based on the quality of the manuscript I suggest it to be accepted after major revision and very careful text editing.
Author Response
Review 1
The article titled "Brazil nut semi-defatted flour oil: Impact of extraction using pressurized solvents on lipid profile, bioactive compounds composition, and oxidative stability” discusses effect of extraction method on lipid profile, bioactive compounds, and stability of oils obtained from the by-product of the Brazilian nut kernel oil pressing process. However interesting the work is, it is not without some shortcomings. In my opinion, the article is carelessly written and needs significant improvements.
Thank you very much for the suggestions. We make the requested corrections, but we are available to improve the article if necessary.
Some remarks:
- Authors are suggested to include a more comprehensive literature review by including recent studies and advancements in the field oil extraction from BNKO.
Answer: The introduction, lines 48 to 57, describes advances in BNK oil extraction.
- In the introduction, the authors mention that BNKO are rich in selenium, magnesium and copper, among others. Why was the effect of the extraction process on the content of these micronutrients not investigated?
Answer: Thank you for your comment. In the introduction section, we mentioned that these micronutrients are presented in the Brazilian nut, which justifies its high consumption and utilization. However, they will not be presented in the Brazilian nut oil. Therefore, we did not evaluate these compounds.
- Inconsistent citation style e.g L78, L80, L134, l327
Answer: The style citation has been revised and corrected. Lines 89, 91, 141 and 347.
- The authors use a lot of abbreviations. They seem to get lost in this and sometimes use different versions of the same abbreviation e.g (BNSFSO=BNSFFO=BNSDFFO?? Or EPF=OPF??). This needs to be unified and corrected.
Answer: The authors are sorry for the incorrect use of abbreviations on some occasions throughout the text. The abbreviation EPF has been replaced by OPF, and the abbreviations BNSFSO, BNSFFO, and BNSDFFO have been replaced by BNSDFO.
- L 15: pressure should be in MPa rather than mPa.
Answer: Corrections were made throughout the text
- L 25: abbreviation used in abstract should be explained.
Answer: The correction was made in the abstract (L. 14).
- L58-65: It is not clear from the quoted part of the article what the novelty of the research presented is. It should be clearly stated.
Answer: The objective of our manuscript extends beyond chemical and physicochemical analyses. The extraction of edible oils using organic solvents, such as hexane, is common in the food and pharmaceutical industries. However, recent studies indicate that impurities in hexane, particularly benzene, pose toxicological risks to the nervous and reproductive systems.
As proposed in our manuscript, the advancement of extraction techniques using pressurized fluids is a significant step towards eco-friendly methods with various industrial applications. This method mitigates the negative impacts of conventional methods on the environment and human health and allows for full solvent recycling and the extraction of solvent-free oil.
Mechanical cold pressing of Brazil nut oil does not allow for complete oil extraction, resulting in semi-defatted flour with low market value due to high lipid oxidation. Pressurized fluids to extract the residual oil from this flour significantly reduce the oil content and enable the extraction of high-purity bioactive compounds without organic solvent residues. The proposed methodology for extracting oil and bioactive compounds from semi-defatted flour is novel, and the results demonstrate its technical feasibility.
Lines 66-75 have been rewritten to highlight the novelty of the work.
- L71: BNSFSO abbreviation should be explained. Does this mean the same thing as BNSFFO? (e. g. L70)
Answer: Yes, the abbreviation has the same meaning, and has been unified in the text.
- L71: In the abstract and section 3.2, the authors state that the experiments were performed at 12 MPa. It should be standardised.
Answer: The information has been standardized throughout the text.
- L71: In Table 1, the authors use the abbreviations EPF[...] and in the text the abbreviations OPF[...]. Is this correct? If so, it should be explained in detail. If not, it should be corrected throughout the text, as this inconsistency is repeated many times. I must admit that this is very disruptive to the interpretation of the research.
Answer: The authors are sorry for the incorrect use of abbreviations on some occasions, and in fact, there are several throughout the text.
- L74-L76: Does this paragraph refer to the explanation of the abbreviations in Table 1? It should be in smaller font to make it clear.
Answer: The requested change has been made. Lines 83-86.
- L101: Table 2. Same situation with abbreviations as in Table 1. OP=BNKO? It is not clear for me.
Answer: The authors are sorry for the incorrect use of abbreviations on some occasions, and in fact, there are several throughout the text.
- L103-105: Repeated unnecessary explanation from L74-L76. This explanation creates more confusion than it clarifies.
Answer: The subtitles have only been retained in Table 1. We kept only the needed information in the footnotes.
- L149-L150: The same situation as in the L103-L105.
Answer: The subtitles have only been retained in Table 1. We kept only the needed information in the footnotes.
- L104: Table 4 is very unreadable. It should be improved.
Answer: Table 4 has been revised for better reading and visualization
- L205-L206: The same situation as in the L103-L105.
Answer: The subtitles have only been retained in Table 1. We kept only the needed information in the footnotes.
- L210: Results for OPF[p1] are not presented in Table 4.
Thanks for the observation. We made the corrections and inserted the results for all extractions.
- L221-L227: Why are the results of the oxidative stability tests not presented in the form of a table or figure?
Answer: The results of the oxidative stability have been altered and are presented in the paper in Table 5.
- L240: BNSDFFO is used in the same meaning as BNSFFO?
Answer: Yes. All abbreviations have been revised and unified in the text.
Based on the quality of the manuscript I suggest it to be accepted after major revision and very careful text editing.
Reviewer 2 Report
Comments and Suggestions for Authors
The manuscript “Brazil Nut Semi-Defatted Flour Oil: Impact of Extraction using Pressurized Solvents on Lipid Profile, Bioactive Compounds Composition, and Oxidative Stability” in my opinion, e.g. European Journal of Lipid Science and Technology or Journal of the American Oil Chemists' Society, is out of the Plants journal scope. The present topic is much more suitable to the journals with oil, technology and extraction, than the plants concept.
The other comments:
- The presented topic is quite well know please see for instance:
https://doi.org/10.1590/S0103-50532008000700021
https://doi.org/10.1016/j.foodres.2010.04.025
https://doi.org/10.1007/978-3-030-30182-8_15
https://doi.org/10.1080/09637480600768077
- The case of beta-tocopherol. I am quite septic about of this type of tocopherol. Othr literature reported that in Brazil nut oil is 82.99 mg of alpha-tocopherol and 116.29 mg gamma-tocopherol [https://doi.org/10.1080/09637480600768077]. Similar observation was reported in another study in Brazil nuts [https://doi.org/10.1016/j.foodres.2010.04.025]. Beta-tocopherol is a rare tocopherol in nature, please see:
https://doi.org/10.3390/molecules27196560
- The overall scientific quality is quite poor.
- Standard deviation for fatty acids max is 0.1. It seems that it is injection of the same sample, rather the experiment repetition.
Comments on the Quality of English Languageno
Author Response
Review 2
The manuscript “Brazil Nut Semi-Defatted Flour Oil: Impact of Extraction using Pressurized Solvents on Lipid Profile, Bioactive Compounds Composition, and Oxidative Stability” in my opinion, e.g. European Journal of Lipid Science and Technology or Journal of the American Oil Chemists' Society, is out of the Plants journal scope. The present topic is much more suitable to the journals with oil, technology and extraction, than the plants concept.
Dear reviewer, thank you for the suggestions. The article submission to "Plants" was for the special issue: "Green Chemistry for Natural Product Extraction: Cleaner and Efficient Approaches". Therefore, we think it fits the journal.
We also revised the English to understand the text better, as requested by the reviewers.
The other comments:
- The presented topic is quite well know please see for instance:
https://doi.org/10.1590/S0103-50532008000700021
https://doi.org/10.1016/j.foodres.2010.04.025
https://doi.org/10.1007/978-3-030-30182-8_15
https://doi.org/10.1080/09637480600768077
Answer: The objective of our manuscript extends beyond chemical and physicochemical analyses. The extraction of edible oils using organic solvents, such as hexane, is common in the food and pharmaceutical industries. However, recent studies indicate that impurities in hexane, particularly benzene, pose toxicological risks to the nervous and reproductive systems.
As proposed in our manuscript, the advancement of extraction techniques using pressurized fluids is a significant step towards eco-friendly methods with various industrial applications. This method mitigates the negative impacts of conventional methods on the environment and human health and allows for full solvent recycling and the extraction of solvent-free oil.
Mechanical cold pressing of Brazil nut oil does not allow for complete oil extraction, resulting in semi-defatted flour with low market value due to high lipid oxidation. Pressurized fluids to extract the residual oil from this flour significantly reduce the oil content and enable the extraction of high-purity bioactive compounds without organic solvent residues. The proposed methodology for extracting oil and bioactive compounds from semi-defatted flour is novel, and the results demonstrate its technical feasibility.
Lines 66-75 have been rewritten to highlight the novelty of the work.
- The case of beta-tocopherol. I am quite septic about of this type of tocopherol. Othr literature reported that in Brazil nut oil is 82.99 mg of alpha-tocopherol and 116.29 mg gamma-tocopherol [https://doi.org/10.1080/09637480600768077]. Similar observation was reported in another study in Brazil nuts [https://doi.org/10.1016/j.foodres.2010.04.025]. Beta-tocopherol is a rare tocopherol in nature, please see:
https://doi.org/10.3390/molecules27196560
Answer: β-tocopherol is found in oil seeds such as sunflower, soybean, and olive (Kamal-Eldin and Roger Andersson, 1997). Several authors have identified the presence of tocopherols in Brazil nuts, including β-tocopherol (Cornelio-Santiago et al., 2019, Chunhieng et al., 2008, Kornsteiner et al., 2006). Results similar to those identified in semi-fatted Brazil nut flour and Brazil nut oil were found by Kornsteiner et al., 2006 (13.2 mg/100g of extracted oil) in Brazil nuts. This study found an average of 15 mg/100 g of oil, a result that is plausible with other studies that have quantified bioactive compounds in Brazil nuts. According to Chunhieng et al., 2008, β-tocopherol is a characteristic component of Brazil nut oil.
- The overall scientific quality is quite poor.
Answer: The Brazil nut is considered one of the most important non-timber forest products globally, producing 28,000 tons in 2022/2023. It is rich in micronutrients such as selenium, one of the foods with the highest selenium content in the world. The Brazil nut tree (Bertholletia excelsa S.B.H.) originates from the Brazilian Amazon. It is one of Earth's most biodiverse and ecologically important regions (Moraes et al., 2021; Venticinque et al., 2016). Deforestation rates in the Brazilian Amazon are the highest among Amazonian countries (Smith et al., 2021). Brazil nut extraction is carried out within protected areas and in traditional communities and is one of the main sources of income and subsistence aid for these communities. Brazil nut oil, as are the nuts, is sold in Brazil and abroad, with low added value. The work aims to improve the extraction yield through pressurized fluid extraction techniques, recovering the remaining oil from a co-product of the pressing extraction process with a high nutritional content, using non-toxic solvents.
- Standard deviation for fatty acids max is 0.1. It seems that it is injection of the same sample, rather the experiment repetition.
Answer - The experiments were duplicated, so the standard deviation is related to true repetitions. The standard deviation is low due to the good replicability and accuracy of the extractors and the gas chromatography used.
Reviewer 3 Report
Comments and Suggestions for Authors
Generally, the paper is well written. However, in my opinion, some of the complaints are as follows:
(1) Table 1, p. 2: Why the authors didn’t give the oil yield obtained experimentally but instead cited 30-40 wt% by Zanqui et al., 2020?
(2) The GC/MS and UHPLC-ESI-MS chromatograms of the (OPF(p2), 4 MPa) should be given.
(3) The word “Flavanonoids” in Table 4 should be changed to “Flavonoids”
(4) Part 3.3.1. Fatty acid profile, active compounds, and antioxidant activity should be divided in the following parts:
-Fatty acids, phytosterols and and tocopherols analysis by GC/MS
-Determination of total phenolic content
-UHPLC-ESI-MS profile of phenolic compounds and
-DPPH assay
Therefore, in my opinion, the manuscript could be accepted after a minor revision.

Author Response
Review 3
Generally, the paper is well written. However, in my opinion, some of the complaints are as follows:
Dear reviewer
Thank you very much for the suggestions. We make the requested corrections, but we are available to improve the articles if necessary.
- Table 1, p. 2: Why the authors didn’t give the oil yield obtained experimentally but instead cited 30-40 wt% by Zanqui et al., 2020?
Answer: Thanks for the observation. In the first version of the article, cold pressing performance was not calculated, so we used the value from the literature. However, in response to the reviewer's request, the experiment was carried out, the yield was determined, and it is included in Table 1.
- The GC/MS and UHPLC-ESI-MS chromatograms of the (OPF(p2), 4 MPa) should be given.
Thanks for the observation. We made the corrections and inserted the results for all extractions.
- The word “Flavanonoids” in Table 4 should be changed to “Flavonoids”
Answer: The change was made.
- Part 3.3.1. Fatty acid profile, active compounds, and antioxidant activity should be divided in the following parts:
-Fatty acids, phytosterols and and tocopherols analysis by GC/MS
-Determination of total phenolic content
-UHPLC-ESI-MS profile of phenolic compounds
-DPPH assay
Answer: The changes were made.
Reviewer 4 Report
Comments and Suggestions for Authors
Many names of identified compounds are given wrong, in the Tables, especially (e.g. Table 2.: use first big letter for Tocopherol; Table 4.: Hydroxybencoi c acid, Hydroxycinna mic acids, etc.).
The names of journals in the part References are given with full- or short-ones.
Comments on the Quality of English LanguageAn English speaking person should check the manuscript before a new submission.
Author Response
Review 4
Dear reviewer
Thank you very much for the suggestions. We make the requested corrections, but we are available to improve the articles if necessary.
We also revised the English to understand the text better, as requested by the reviewers.
- Many names of identified compounds are given wrong, in the Tables, especially (e.g. Table 2.: use first big letter for Tocopherol; Table 4.: Hydroxybencoi c acid, Hydroxycinna mic acids, etc.).
Answer: Table 4 has been revised, and the names of the compounds have been corrected.
- The names of journals in the part References are given with full- or short-ones.
Answer: The journals' names in the references section have been revised and updated to full names.
Round 2
Reviewer 1 Report
Comments and Suggestions for Authors
Accept in present form
Author Response
Dear Reviewer
Thank you very much for considering our work.
Kind regards
Reviewer 2 Report
Comments and Suggestions for Authors
I still keep my previous decision. I think that scientific quality is quite poor which is confirmed by the answer to the question associated with the beta-tocopherol. Yes indeed, beta-tocopherol can be found in seed oils such as sunflower, soybean, olive, and other oils as well as can be seen in the reference provided below,
https://doi.org/10.1016/j.foodres.2022.112386
however as a minor tocopherol, not a major. The authors should scientificaly (experimental work) to confirm the dominance of beta-tocopherol in Brazil nuts. The authors do not provide the sources of tocopherol standards, so I suppose any standards were not used. So I wonder how the authors know that is beta-tocopherol, not gamma-tocopherol, while both of them are isomers and have the same molecular mass. Can you explain?
non
Author Response
Author's Reply to the Review Report (Reviewer 2)
- Yes indeed, beta-tocopherol can be found in seed oils such as sunflower, soybean, olive, and other oils as well as can be seen in the reference provided below, however as a minor tocopherol, not a major.
Answer: We agree with the reference indicated by the reviewer, who consistently clarifies the similarity in structure and properties between the isomers of the alpha-tocopherol, beta-tocopherol, and gamma-tocopherol molecules. This similarity makes it difficult to differentiate between these isomers without an appropriate methodology.
- The authors should scientificaly (experimental work) to confirm the dominance of beta-tocopherol in Brazil nuts. The authors do not provide the sources of tocopherol standards.
Answer: As described in section 3.3.1, the phytosterols and tocopherols present in the samples were derivatized with N, O-Bis(trimethylsilyl)trifluoroacetamide with trimethylchlorosilane (BSTFA/TMCS) at 60 °C for 30 min. 80 μL of the 5-α-cholestane standard at 5 mg mL-1 were added to the derivatized samples and diluted with heptane to a final 40 mg mL-1 concentration. The diluted samples were injected onto an SH-RTx-5MS capillary column (Shimadzu, 30 m × 0.25 mm × 0.25 μm), following a heating ramp identical to that used for other vegetable oils [13,17,30,38].
- So I wonder how the authors know that is beta-tocopherol, not gamma-tocopherol, while both of them are isomers and have the same molecular mass. Can you explain?
Answer: The information regarding the content of β-tocopherol and γ-tocopherol was obtained using the methodology described in item 3.3.1 and with the aid of the available mass spectrometry database. Based on the reference indicated by the reviewer, we believe that the best way to express the tocopherol content obtained in the samples analyzed is to express it as a racemic mixture of tocopherols. Therefore, the term referring to β-tocopherol in the manuscript will be changed to racemic mixture of tocopherols.
Reviewer 4 Report
Comments and Suggestions for Authors
1 minor mistake at the part "References" was found, as mentioned.
Author Response
Dear reviewer
Thank you very much for the observation.
We have reviewed all references and believe that they are now all correct.
Kind regards
Round 3
Reviewer 2 Report
Comments and Suggestions for Authors
Scientific quality is low. Lack of sufficient scientific explanations. Isomers of tocopherols are only two beta-tocopherol and gamma-tocopherol. When we talk about alpha, beta, gamma, and delta we say homologues. The lack of using standards.
"racemic mixture tocopherol" - what this mean? beta and gamma? or alpha, beta, gamma, and delta? It was used GC-MS so the situation should be clear. For phenolics was provided detailed information about the identification was, while in the case of squalene, β-sitosterol, tocopherols not, and lack of the standards.
"Scalene" - I think you mean "squalene"?
How fatty acid can be expressed in g/100 g oil, without using standards? Without using standards, you can only do identification.
Author Response
Author's Reply to the Review Report (Reviewer 2)
Dear reviewer
We apologize for the unclear information and greatly appreciate your comments. We reviewed the methodological part and results and hope they are adequately explained.
- Isomers of tocopherols are only two beta-tocopherol and gamma-tocopherol. When we talk about alpha, beta, gamma, and delta we say homologues. The lack of using standards. "racemic mixture tocopherol" - what this mean? beta and gamma? or alpha, beta, gamma, and delta? It was used GC-MS so the situation should be clear. For phenolics was provided detailed information about the identification was, while in the case of squalene, β-sitosterol, tocopherols not, and lack of the standards.
Answer: The term racemic mixture tocopherol was deleted, and the values ​​were expressed as β+γ-tocopherol. For identification, the mass spectra of the compounds detected were compared with those of the NIST 14 library, the Pubchem database established by the National Library of Medicine, and the NIST standard reference database number 69 (NIST Chemistry WebBook).
- "Scalene" - I think you mean "squalene"?
Answer: The correction has been made.
- How fatty acid can be expressed in g/100 g oil, without using standards? Without using standards, you can only do identification."
Answer: The percentage of each compound was determined using the area normalization method, defined as the ratio between the area of the individual compound by the total area of the peaks obtained in the chromatogram. Thus, the unit g/100 g oil was changed to % in Table 2.
The mass spectra of the compounds detected were compared with those of the NIST 14 library, the Pubchem database established by the National Library of Medicine, and the NIST standard reference database number 69 (NIST Chemistry WebBook).
Round 4
Reviewer 2 Report
Comments and Suggestions for Authors
Some improvements have been made, but ...
They were missing how squalene, tocopherols, and beta-sterol were calculated.
The issue of fatty acids, yes, using MS you can make an identification of compounds but their percentage expression by MS is not quite correct because the MS detector for each molecule reacts more or less intensively. This causes the signal for 1 g of compound "x" to be close to 100 g of compound "y".
There are still many minor errors in the manuscript such as spelling separately and together "7.5%" vs "2.2%".
"We demonstrated that this oil has high-quality attributes, promoting a more sustainable and economically viable approach to the industry. " - I disagree with this statement.
" In conclusion, oils with better fatty acid profiles, phytosterol composition, and suitable antioxidant activity may be obtained using pressurized fluids" - I disagree with this statement. What do you mean by better?
"The mixture of solvents may improve the concentration of squalene, while using only n-propane may increase oil yield." - what do you mean by "mixture of solvents may improve the concentration"? Do you mean recovery?
The manuscript would gain in value once the language is improved.
Author Response
Dear Reviewer
Thank you once again for your observations and suggestions on our work. We made the necessary modifications and hope it is better.
- They were missing how squalene, tocopherols, and beta-sterol were calculated.
Answer: To quantify the content of each compound, the internal standardization technique was used, using 5-α-cholestane as an internal standard.
- There are still minor errors in the manuscript such as spelling separately and together "7.5%" vs "2.2%.
Answer: The correction was made as indicated by the reviewer.
- We demonstrated that this oil has high-quality attributes, promoting a more sustainable and economically viable approach to the industry. " - I disagree with this statement. In conclusion, oils with better fatty acid profiles, phytosterol composition, and suitable antioxidant activity may be obtained using pressurized fluids" - I disagree with this statement. What do you mean by better?"The mixture of solvents may improve the concentration of squalene, while using only n-propane may increase oil yield." - what do you mean by "mixture of solvents may improve the concentration"? Do you mean recovery?
Answer: We agree. The manuscript text in the conclusion item was changed.
Round 5
Reviewer 2 Report
Comments and Suggestions for Authors
There is still the issue of English e.g. "To quantify the content of each compound, the internal standardization technique was used, using 5-α-cholestane as an internal standard."
I already mentioned that "O" after numbers in names of compounds such as "myricetin 3-o-rhamnoside" should be written in italic style.
The information for the identification of squalene, tocopherols, and beta-sterol should be provided the same as it was done for phenolic compounds.
Line 258 "n-propane" should be started with a big letter "P" and italic style for "n".
I have some doubts about the DPPH assay. "For DPPH•, an aliquot of the ethanolic solution of the oil ..." - oils are not well dissolved in ethanol. In 2-propanol and ethyl acetate yes and then could be mixed with the ethanol solution of DPPH•. This part needs to be described in much more detail since the used procedure was not the correct one.
The conclusion part is still badly written.
Language style "In conclusion, the pressurized fluid extraction technology applied to Brazil nuts and semi-fatted Brazil nut flour has yielded oils with higher concentrations of linoleic acid and β-sitosterol than cold-press extraction."
"higher concentrations of linoleic acid" - how you can write higher concentrations in the case of fatty acid which was not even determined quantitatively just by MS detector % which is not correct in my opinion. The % is correct only in the case of the FID detector.
In general, the whole manuscript should be checked by a professional person to catch any other potential mistakes and to improve the language style.
non